# Structural Colored Fabric Based on Monodisperse Cu_2_O Microspheres

**DOI:** 10.3390/ma17133238

**Published:** 2024-07-01

**Authors:** Xiaowen Li, Zhen Yin, Zhanghan She, Yan Wang, Parpiev Khabibulla, Juramirza Kayumov, Guojin Liu, Lan Zhou, Guocheng Zhu

**Affiliations:** 1Key Laboratory of Advanced Textile Materials and Manufacturing Technology, Ministry of Education, Zhejiang Sci-Tech University, Hangzhou 310018, China; lxw5201019@163.com (X.L.); 19895921166@163.com (Z.S.); 15757157466@163.com (G.L.); lan_zhou330@163.com (L.Z.); 2College of Textile Science and Engineering, Zhejiang Sci-Tech University, Hangzhou 310018, China; 17712107935@163.com (Z.Y.); amywang1021@hotmail.com (Y.W.); 3Zhejiang-Czech Joint Laboratory of Advanced Fiber Materials, Zhejiang Sci-Tech University, Hangzhou 310018, China; 4Department of Technology of Textile Industry Products, Namangan Institute of Engineering and Technology, 7, Kasansay Street, Namangan 160115, Uzbekistan; parhabib@mail.ru; 5Department of Civil Engineering, Samarkand State Architecture and Construction University, Samarkand 140143, Uzbekistan; juramirza@gmail.com; 6Zhejiang Provincial Innovation Center of Advanced Textile Technology, Shaoxing 312000, China

**Keywords:** Cu_2_O microsphere, photonic crystal, structural color, textile

## Abstract

Structural-colored fabrics have been attracting much attention due to their eco-friendliness, dyelessness, and anti-fading properties. Monodisperse microspheres of metal, metal oxide, and semiconductors are promising materials for creating photonic crystals and structural colors owing to their high refractive indices. Herein, Cu_2_O microspheres were prepared by a two-step reduction method at room temperature; the size of Cu_2_O microspheres was controlled by changing the molar ratio of citrate to Cu^2+^; and the size of Cu_2_O microspheres was tuned from 275 nm to 190 nm. The Cu_2_O microsphere dispersions were prepared with the monodispersity of Cu_2_O microspheres. Furthermore, the effect of the concentration of Cu_2_O microsphere and poly(butyl acrylate) on the structural color was also evaluated. Finally, the stability of the structural color against friction and bending was also tested. The results demonstrated that the different structural colors of fabrics were achieved by adjusting the size of the Cu_2_O microsphere, and the color fastness of the structural color was improved by using poly(butyl acrylate) as the adhesive.

## 1. Introduction

Structural color is generated from the interaction of light and microstructures without any role having to be played by dyes or pigments [1,2]. It widely exists in nature, has been researched for a long time, and has attracted much attention due to its advantages of eco-friendliness, brilliant colors, and anti-fading properties [3,4]. The fundamental optical processes of structural color can be defined as thin-film interference, multilayer interference, a diffraction grating effect, light scattering, and photonic crystals [4,5,6,7,8]. Among them, the most commonly applied methods explored are multilayered photonic crystals [9,10,11,12,13] or amorphous photonic structures [14,15,16,17] for the structural coloration of textiles.

The widely used colloidal spheres for constructing photonic crystals are polystyrene, polymethyl methacrylate, polysulfide and silica spheres [18,19,20,21], which have a relatively low refractive index, resulting in a weak band gap for the photonic crystal and a less brilliant structural color [22]. Some researchers reported that a high refractive index would be helpful for generating brilliant structural color, and metallic microspheres are usually selected [23,24]. Bi [23,24] synthesized Cu_2_O microspheres at room temperature for structural color formation, and distinct colors were obtained on transparent substrates through the structural arrangement of these Cu_2_O microspheres. For instance, Han [25] sprayed Cu_2_O microspheres of various sizes onto textiles to avoid iridescence and achieve better mechanical stability. Sarwar [26] synthesized cross-linkable nano-cubes of Cu_2_O and used them to color and functionalize cotton fabrics simultaneously to obtain self-cleaning, antibacterial fabric based on photocatalytic oxidation reactions. This strategy serves as a potential alternative to toxic chemical dyeing and finishing. The exploration of microspheres of Cu_2_O to achieve structural coloration using the spray coating method along with functional properties was extended by Fang [27], in which they finely tuned the size of Cu_2_O particles explored the structural color stability and fastness along with their antibacterial properties.

However, there is an essential need to explore the full potential of these photonic crystal structures, i.e., Cu_2_O microspheres as building blocks for structural coloration of textiles, the stability of photonic crystals, and the color fastness of structurally colored textiles. Herein, in this research, Cu_2_O microspheres in different sizes by a two-step reduction method are proposed and developed, followed by the formulation of Cu_2_O microsphere dispersions. The morphology, monodispersity, microstructure, and stability of these Cu_2_O microsphere dispersions and the obtained structurally colored fabrics have been assessed in detail. This research would help understand the structurally colored fabrics produced by Cu_2_O microspheres in detail and would inspire researchers to develop new technologies.

## 2. Materials and Methods

### 2.1. Materials

Copper acetate monohydrate (C_4_H_6_CuO_4_·H_2_O), sodium citrate dehydrate (C_6_H_5_Na_3_O_7_·2H_2_O), polyvinylpyrrolidone (MW = 58,000), ethylene glycol, ascorbic acid, sodium hydroxide, anhydrous ethanol, deionized water (with electrical conductivity 18 MΩ·cm) were provided by Shanghai Macklin Biochemical Technology Co., Ltd. (Shanghai, China). The woven polyester fabrics were obtained from Suzhou Feijun Textile Co., Ltd. (Suzhou, China). Poly(butyl acrylate) (MW = 210,000) was purchased from Dayang Chem (Hangzhou) Co., Ltd. (Hangzhou, China). All the chemicals and solvents were analytical grade and were used as received without further purification.

### 2.2. Preparation of Cu_2_O Microsphere

The Cu_2_O microspheres were prepared by a two-step reduction method at room temperature. Specifically, 0.32 g of copper acetate monohydrate, 0.52 g sodium citrate, and 1.50 g polyvinylpyrrolidone were dissolved in a mixed solvent of 30 g ethylene glycol and 100 g deionized water to form a homogenous solution under stirring for 20 min at 25 °C. Then, 20 mL of sodium hydroxide solution (0.5 M) was added to the homogenous solution at room temperature with stirring to form a new solution named S1.

A transparent ascorbic acid solution was prepared by dissolving 0.32 g of ascorbic acid in 15 mL of deionized water. A total of 7.5 mL of the ascorbic acid solution was added into solution S1 at a speed of 1 drop per second by using the injection pump. After 5 min, the remaining 7.5 mL of ascorbic acid solution was simultaneously added in the above system under stirring for 60 min to get the solution named S2.

A high-speed centrifuge (TGL-20B-C) from Shanghai Anting Scientific Instrument Factory (Shanghai, China) was used to separate Cu_2_O microspheres from solution S2 at a speed of 8000 r/m for 30 min. The as-prepared Cu_2_O microspheres were collected and put into an ethanol-H_2_O mixture solution, and then the Cu_2_O microspheres/ethanol-H_2_O solution was separated again in the high-speed centrifuge. After repeating this process three times, the separated Cu_2_O microspheres were then dried in a vacuum at 60 °C for 3 h. The different sizes of Cu_2_O microspheres were achieved by changing the molar ratio of citrate to Cu^2+^. The molar ratio of citrate to Cu^2+^ was 0.8, 1, 1.1 and 1.2.

### 2.3. Preparation of Structural Color Ink

The monodisperse Cu_2_O microspheres were added to anhydrous ethanol, and the mixture was stirred for 30 min to obtain a stable ethanol dispersion of the Cu_2_O microsphere. The concentration of Cu_2_O microsphere in the mixture dispersion was 0.5%, 1%, 3% and 5%. Then, the poly(butyl acrylate) as the adhesive was added into the mixture dispersion under stirring to form a homogenous solution, i.e., structural color ink. The concentration of poly(butyl acrylate) was 5%, 10%, 15%, 20% and 25%.

### 2.4. Requirement of Cu_2_O Microsphere

The mechanism of structural color produced by photonic crystal follows the Bragg diffraction law in optics, which can be described as Equation (1).
(1)$${m\lambda }_{\max}=2d_{\mathrm{hkl}}{\left(n_{\mathrm{avg}}^{2}\;-\;\sin^{2}\theta \right)}^{1/2}$$
where *m* is the order of diffraction, *λ_max_* is the wavelength corresponding to the maximum peak in the reflected spectral line (i.e., the position of the photonic bandgap), *d_hkl_* is the interplanar spacing, *n_avg_* is the average refractive index of a three-dimensional array, *θ* is the angle of incidence of incident light. The interplanar spacing of *d_hkl_* can be calculated by Equation (2).
(2)$$d_{\mathrm{hkl}}=\frac{\sqrt{2}D}{\sqrt{h^{2}+k^{2}+l^{2}}}$$
where *D* is the average diameter of microspheres, *h*, *k*, and *l* represents the crystal plane index in the crystal structure.

The refractive index of Cu_2_O is 2.7 [28], therefore the range of the Cu_2_O microsphere’s diameter should be between 168 nm and 278 nm for the visible light wavelength range [29]. Additionally, when the monodisperse index (PDI) of the microsphere is less than 0.08 (PDI ≤ 0.08), the Cu_2_O dispersion would have excellent monodispersity, which is helpful for the construction of a photonic crystal [30].

### 2.5. Characterization Methods

Dynamic Light Scattering and Zeta Potential measurement systems Zetasizer Nano S from Malvern Panalytical (Malvern, UK) was applied to evaluate the size, and the PDI of the Cu_2_O microsphere can be obtained according to Equation (3).
(3)PDI=σ2ZD2
where *PDI* is a monodisperse index, *σ* is the standard deviation, and *Z_D_* is the hydrated size of the Cu_2_O microsphere. The monodispersity of Cu_2_O microsphere dispersion is inversely proportional to the PDI.

Specifically, the Cu_2_O microsphere dispersion was diluted into transparency, and then the diluted dispersion was put into a quartz cell for testing. The average values were taken from three measurements. The surface morphology of the Cu_2_O microsphere was determined using a field emission scanning electron microscope (FESEM, ALTRA55, Carl Zeiss SMT Ltd, Oberkochen, Germany). Fourier transform infrared spectroscopy (FT-IR) analyses were performed to determine the chemical compounds in the obtained Cu_2_O microsphere using a Thermo Scientific (Waltham, MA, USA) Nicolet 5700 FT-IR spectrometer with the KBr method. The Cu_2_O microsphere and KBr were mixed at a ratio of 1:50, and then the mixture was ground into powder and pressed into a transparent film for testing. The scanning range was 4000–600 cm^−1^, with a scanning resolution of 4 cm^−1^ and a scanning frequency of 32 times. X-ray photoelectron spectroscopy XPS, Thermo Scientific K-Alpha (Waltham, MA, USA) was used to determine the element valence of the Cu_2_O microsphere, the analysis of the energy spectrum is based on the binding energy of the C1s peak at 284.6 eV. The crystal phase structures of the Cu_2_O microsphere were analyzed by an X-ray diffractometer (XRD, K-alpha, Thermo Fisher, Oxford, UK). The dried Cu_2_O microspheres were ground into powder as a sample, the emission source was CuK_α_ ray (wavelength was 0.154178 nm), the scanning speed was 10 °/min, the scanning angle was 10~80°, the electric current and electric voltage were 40 mA and 40 kV, respectively.

### 2.6. Storage Stability of Cu_2_O Microsphere Dispersion

Zeta potential values of Cu_2_O microsphere dispersions with different sizes of Cu_2_O microsphere were evaluated by a Zeta potential analyzer (Brook-21) from Brookhaven Instruments, Holtsville, NY, USA. The average value was taken from three measurements. The Cu_2_O microsphere dispersions were sealed in a polyethylene terephthalate bottle and stored at room temperature (25 °C). After a period of time, the Cu_2_O microsphere dispersion was observed to determine whether it was layering by a digital camera. Meanwhile, the diameter and monodispersity of Cu_2_O microspheres were analyzed by a dynamic light scattering particle size distribution analyzer to determine their storage stability.

### 2.7. Characterization of Structural Color and Its Color Fastness

The structural color generated by Cu_2_O microsphere photonic crystal was observed by a 3D video microscope (KH-7700 from QUESTAR China Limited, Shanghai, China). The refractivity of structural color was evaluated by UV-VIS spectrophotometer Lambda-35 by PerkinElmer (Waltham, MA, USA), and the observation angle was perpendicular to the sample (90°).

The bending test: the abatement of structural color and the abscission of photonic crystal were observed before and after the bending test. Specifically, the structural colored fabric with a size of 6 cm × 6 cm was folded in the middle until the two ends were contacted, and then the structural colored fabric was opened. The process was repeated 150 times, and then the fabric was evaluated. The friction test: the rubbing fastness test was carried out by referring to the standard ISO 105-X12:2016 (Textiles—Tests for color fastness Part X12: Color fastness to rubbing) [31]. Specifically, the structural colored fabric was placed and fixed on a flat plane, and then a weight of 50 g was put onto the fabric, the weight was moved on the surface of fabric from one side to another side. The process was repeated for 150 times, and then the fabric was evaluated.

## 3. Results and Discussion

### 3.1. Morphology of Cu_2_O Microsphere

The Cu_2_O microspheres have smooth surfaces, excellent sphericity, and uniform size. Cu_2_O microspheres with four different sizes were synthesized under different conditions, which were 275 nm, 240 nm, 210 nm, and 190 nm, respectively (Figure 1).

### 3.2. Analysis of Monodispersity of Cu_2_O Microsphere

The diameter and monodispersity of the Cu_2_O microsphere were tuned by changing the molar ratio of citrate to Cu^2+^. It was found that the diameter of the Cu_2_O microsphere had a positive correlation with the molar ratio of citrate to Cu^2+^, i.e., the increase in molar ratio led to an increase in Cu_2_O microsphere diameter (Figure 2). The Cu_2_O microsphere had monodispersed indices of less than 0.08 (Figure 2), which indicated that the synthesized Cu_2_O microspheres have excellent stability and monodispersity [30]. Therefore, the synthesized Cu_2_O microspheres would have a suitable dispersion for the construction of photonic crystals.

### 3.3. Microstructure of Cu_2_O Microsphere

The structural composition of Cu_2_O microspheres was analyzed by FTIR. C=O stretching vibration absorption peak was observed at 1632 cm^−1^, this peak is wider than the carbonyl absorption peak of polyvinylpyrrolidone (PVP), which is due to the overlapping of the carboxyl asymmetric stretching vibration peak (1598 cm^−1^) and the carbonyl absorption peak of citrate [32]. The symmetric contraction vibration absorption peak of carboxylic acid in Cu_2_O microspheres was found at 1415 cm^−1^, and the stretching vibration peak of Cu-O was found at 630 cm^−1^; refer to Figure 3. The FTIR analysis indicated that there were PVP and carboxylate ions on the Cu_2_O microsphere.

In order to further understand the purity and composition of the Cu_2_O microsphere, XPS testing was carried out. The Cu 2p spectrum shows a sharp main peak at 932 eV (Figure 4A), which corresponds to Cu^2+^ in Cu_2_O. At 934 eV, there is a very weak peak corresponding to Cu^2+^ in CuO. The appearance of Cu^2+^ in the product is due to the oxidation of the surface part of the microsphere.

The O 1s spectrum of Cu_2_O microspheres also shows two peaks: the peak at 531 eV belongs to O in Cu_2_O (Figure 4B), and the small peak at 530 eV belongs to O in CuO. The appearance of Cu^2+^ in the product is due to the oxidation of the surface part of the microsphere. Although the XPS spectrum confirms the presence of CuO in the product, the content of CuO is relatively small, which may only be due to the oxidation of the surface of the microsphere.

### 3.4. XRD of Cu_2_O Microsphere

In order to determine the purity and crystal structure of Cu_2_O microspheres, XRD tests were conducted on the prepared Cu_2_O microspheres. As shown in Figure 5, the Cu_2_O microspheres are consistent with the standard spectrum of cubic Cu_2_O in PDF No. 05-0667 refer to Figure 5B. From Figure 5A, it can also be seen that there are no impurity peaks in the product, indicating that the Cu_2_O microsphere has a relatively pure crystal phase.

### 3.5. Zeta Potential Analysis of Cu_2_O Microsphere Dispersion

The stability of the Cu_2_O microsphere dispersion can be evaluated by observing the absolute value of the Zeta potential difference between the continuous phase of the particles and the liquid stable layer. Usually, when the absolute value of the Zeta potential of particles in the aqueous phase is greater than or equal to 30 mV, it is considered that the dispersion stability of the system is good [33].

The Zeta potential values of Cu_2_O microsphere dispersions were −31.24 mV, −32.34 mV, −30.85 mV, and −33.43 mV, corresponding to the diameters of Cu_2_O microsphere 275 nm, 240 nm, 210 nm, and 190 nm (the data are given in Table 1). All the absolute Zeta potential values were over 30 mV, indicating that the microspheres are not easy to agglomerate, and the dispersions had good dispersion.

As shown in Figure 6A, as the storage time increased, the particle size of the microspheres showed a stable trend, which indicated that the size of the microspheres exhibited excellent uniformity. However, the monodispersity index PDI of the microsphere dispersions showed a fluctuation trend when the storage time increased, as shown in Figure 6B. But the PDI of all dispersions after storage still did not exceed 0.08, which further proved that the Cu_2_O microspheres had good monodispersity. Therefore, the Cu_2_O microsphere dispersions synthesized in this study meet the requirements for constructing photonic crystal structures.

### 3.6. Effect of Concentration of Cu_2_O Microsphere on Structural Color

The Cu_2_O microspheres obtained by centrifugation were dry particles that were prone to aggregate together and could not be directly applied to the spray preparation of structural color patterns. In order to solve this problem, the Cu_2_O microsphere was added to anhydrous ethanol with different concentrations from 0.5 wt.% to 5.0 wt.%. The Cu_2_O microspheres with a size of 240 nm were successfully applied to fabrics, which demonstrated structural colors; refer to Figure 7. The structural colors changed obviously when the concentration of Cu_2_O microsphere increased from 0.5 wt.% to 5.0 wt.%. When the concentrations of Cu_2_O microsphere were 0.5 wt.% and 1.0 wt.%, though the fabric showed a structural color effect, most of the fabric’s background color could still be observed. Moreover, the structural color effect on the fabric was brighter when the mass fraction of microspheres increased to 3.0 wt.%, while the texture of the fabric can also be seen; refer to Figure 7C. Furthermore, when the mass fraction of microspheres was 5.0 wt.%, the fabric surface became fully covered. Although the structural color can be seen on the fabric, the structural layer became thicker, and the structural color began to turn whiter, as shown in Figure 7D.

The arrangement of Cu_2_O microspheres was observed as well and given in Figure 7E–H. The concentration of Cu_2_O microspheres had an effect on their arrangement. When the concentration of the Cu_2_O microsphere was 0.5 wt.%, the Cu_2_O microsphere was partially regularly arranged, and there were many gaps among the Cu_2_O microsphere (Figure 7E), which led to a structural color with low brightness. Meanwhile, the Cu_2_O microsphere was not well covered with the fabrics, leading to the appearance of fabric (Figure 7A). When the mass fraction was 1.0 wt.%, the concentration of microspheres was still low and the arrangement was irregular (Figure 7F), which had a certain impact on the arrangement of the photonic crystal structure and on the structural color. When the mass fraction of microspheres reached 3.0 wt.%, the arrangement of microspheres became dense and orderly (Figure 7G), thus presenting a bright and brilliant structural color. When the mass fraction of microspheres reached 5.0 wt.%, the arrangement of microspheres was relatively irregular and overlaid (Figure 7H), resulting in a poor structural color effect on the fabric.

It is worth noting that when the concentration of microspheres in the structural color ink was too low, gaps were generated during the assembly process due to the lower number of microspheres, which ultimately affected the tightness of the arrangement of the microspheres. As the mass fraction of microspheres gradually increased, the arrangement of microspheres became regular and uniform. However, when the concentration of Cu_2_O microspheres was too high, a large number of microspheres would collide with each other, thus affecting the formation of the regular self-assembly structure. Based on the above analysis, when the mass fraction of microspheres was 3.0 wt.%, it is possible to establish a photonic crystal chromogenic structure with good integrity and structural color effect on the textile substrate.

### 3.7. Effect of Concentration of Adhesive on Structural Color

Generally, colloidal microsphere structural primitives have brighter structural colors after self-assembly; however, their bonding fastness with the substrate is often poor, which limits their practical application. To improve its mechanical strength, an adhesive is generally added to the system. Although the fastness is improved after the adhesive is added, the structural color effect becomes worse, or a redshift occurs.

The effect of different poly(butyl acrylate) ratios on structural color patterns was investigated by observing the surface morphology and color of fabrics. It can be seen that when the adhesive ratio was between 5.0 wt.% and 10.0 wt.%, the surface of the fabric was completely covered by the photonic crystal structural color, and the fabric texture can be seen, showing a bright structural color (Figure 8A,B). When the ratio of poly(butyl acrylate) exceeded 10.0 wt.%, although the fabric surface also showed structural color, it started to appear slightly whitened (Figure 8C,D). When the ratio of poly(butyl acrylate) was 25.0 wt.%, the photonic crystal structural color almost changed (Figure 8E). The results had the agreement with other research work that the adhesive will change the performance of structural color [34].

### 3.8. Effect of Size of Cu_2_O Microsphere on Structural Color

Using the spraying method, Cu_2_O microsphere structural color ink with different particle sizes was used to construct a structural color pattern on the surface of the textile substrate. The results are given in Figure 9. It can be seen that the structural color pattern integrity and structural color effect constructed were good, and the structural color was tuned by changing the size of Cu_2_O microsphere. When the size of Cu_2_O microsphere increased, the spacing among the Cu_2_O microsphere increased as well, which led to a shifting of the reflected wavelengths to longer wavelengths, i.e., to a redder color [35].

The reflectivity of the constructed structural color patterns was evaluated, and the results are presented in Figure 10. The structural color pattern formed by microspheres of various particle sizes showed a high level of reflectivity and a sharp reflection peak, which means that the structural color effect was good. Meanwhile, the reflection spectra of the structure colors also had a similar trend with the photographs, i.e., the larger size of the Cu_2_O microsphere tended to be a redder color.

### 3.9. Stability of Photonic Crystal

#### 3.9.1. Bending Test

The structural colored fabric for the bending test was prepared by using Cu_2_O microspheres with a diameter of 240 nm; the concentration of Cu_2_O microspheres was 3.0 wt.%, and the concentration of poly(butyl acrylate) was 5.0 wt.%. After the bending test, the structurally colored fabrics still demonstrated the same bright color (Figure 11(A1–A4)). The structures of the Cu_2_O photonic crystal before and after the bending test were also observed by SEM, which showed that the Cu_2_O photonic crystal was orderly arranged and did not have a detachment phenomenon after the bending test (Figure 11(B1,B2)). This could be due to the bonding function of poly(butyl acrylate), which provided a solid interface strength and stable structural color.

#### 3.9.2. Friction Test

The structural colored fabric was prepared by using Cu_2_O microspheres with a diameter of 210 nm; the concentration of Cu_2_O microspheres was 3.0 wt.%, and the concentration of poly(butyl acrylate) was 5.0 wt.%. A friction test was conducted to evaluate the stability of the structural color pattern. The results demonstrated that the structural color was the same color before and after the friction test (Figure 12(A1–A4)), which indicated that the Cu_2_O photonic crystal had a stable structure. The SEM images of Cu_2_O photonic crystal before and after the friction test showed that there was almost no difference in the arrangement of Cu_2_O microspheres and no shedding of Cu_2_O observed (Figure 12(B1,B2)). The results also approved that the poly(butyl acrylate) as adhesive improved the stability of photonic crystals and the color fastness of structurally colored fabrics, indicating excellent durability of structural color.

## 4. Conclusions

Cu_2_O microspheres in different sizes and monodisperse indices have been achieved by controlled synthesis for structural color textiles. The surface morphology, monodispersity, microstructure, and stability of Cu_2_O microsphere dispersions were rigorously assessed. The effect of the molar ratio of citrate to Cu^2+^ on size and monodispersity and the microstructure of Cu_2_O microspheres were also studied in detail. The important parameters, including the composition, purity, and crystal structure to assure the quality of the microspheres of the synthesized Cu_2_O microspheres, were confirmed by FTIR, XPS, and XRD. The effect of the concentration of Cu_2_O microsphere and adhesive on the structural color performance was also assessed. The adhesive strength of the Cu_2_O microspheres and fabric was increased by adding poly(butyl acrylate) as an adhesive. The microspheres were well bound enough so that they could resist frictional and bending forces. This study would be useful to advance and fine-tune the structural coloration of fabrics with better mechanical stability.

## Figures and Tables

**Figure 1 materials-17-03238-f001:**
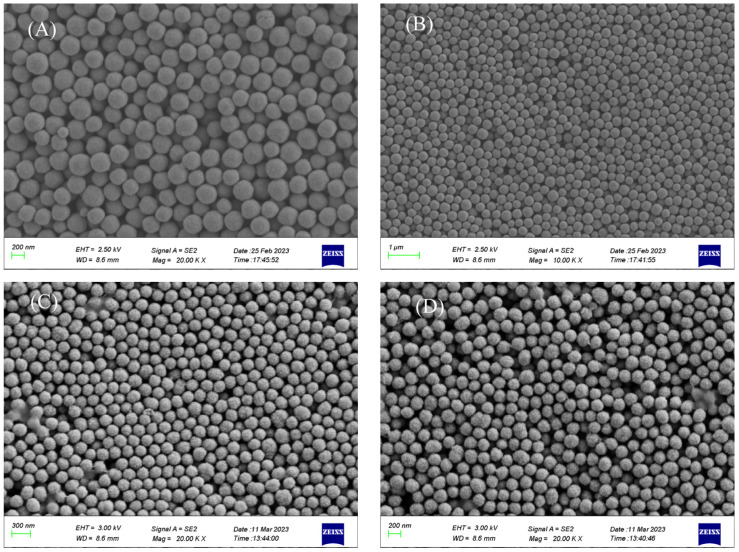
Morphology of Cu_2_O microsphere with different sizes (**A**) 275 nm (**B**) 240 nm (**C**) 210 nm (**D**) 190 nm.

**Figure 2 materials-17-03238-f002:**
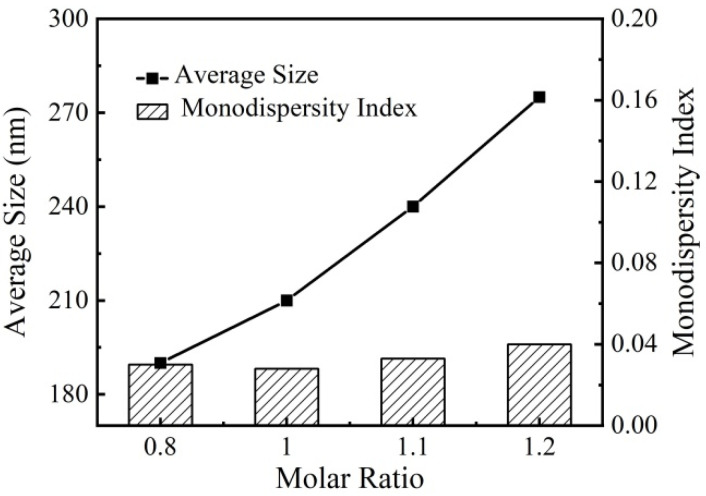
The effect of the molar ratio of citrate to Cu^2+^ on the diameter and monodispersity of the Cu_2_O microsphere.

**Figure 3 materials-17-03238-f003:**
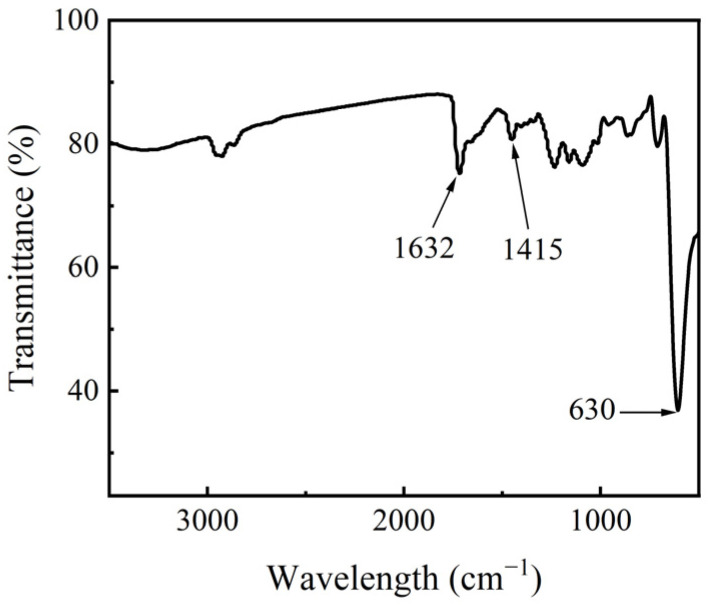
FTIR of Cu_2_O microsphere with a diameter of 275 nm.

**Figure 4 materials-17-03238-f004:**
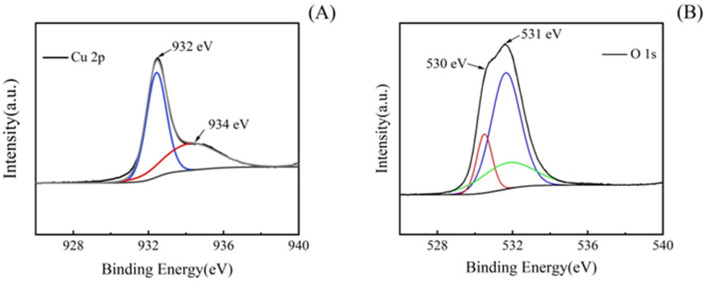
XPS spectrum of Cu_2_O microsphere with a diameter of 275 nm. (**A**) The Cu 2p spectrum of the Cu2O microsphere. (**B**) The O 1s spectrum of the Cu2O microsphere

**Figure 5 materials-17-03238-f005:**
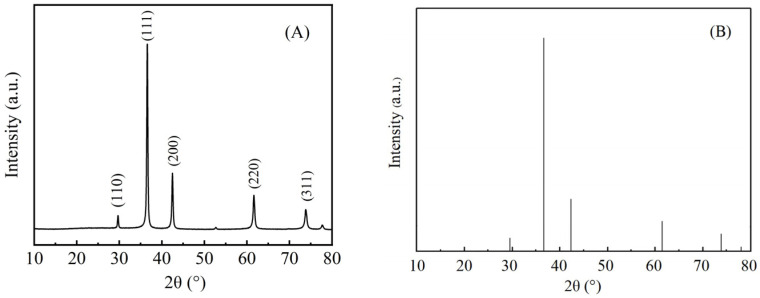
XRD spectrum of (**A**) Cu_2_O microsphere with a diameter of 275 nm, (**B**) Cu_2_O (PDF#05-0667).

**Figure 6 materials-17-03238-f006:**
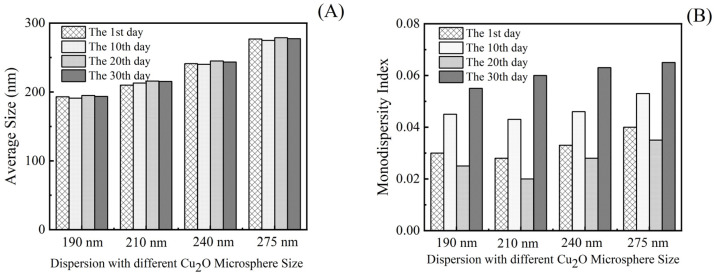
The effect of storage time on (**A**) the diameter of the Cu_2_O microsphere, (**B**) the monodispersity of the Cu_2_O microsphere.

**Figure 7 materials-17-03238-f007:**
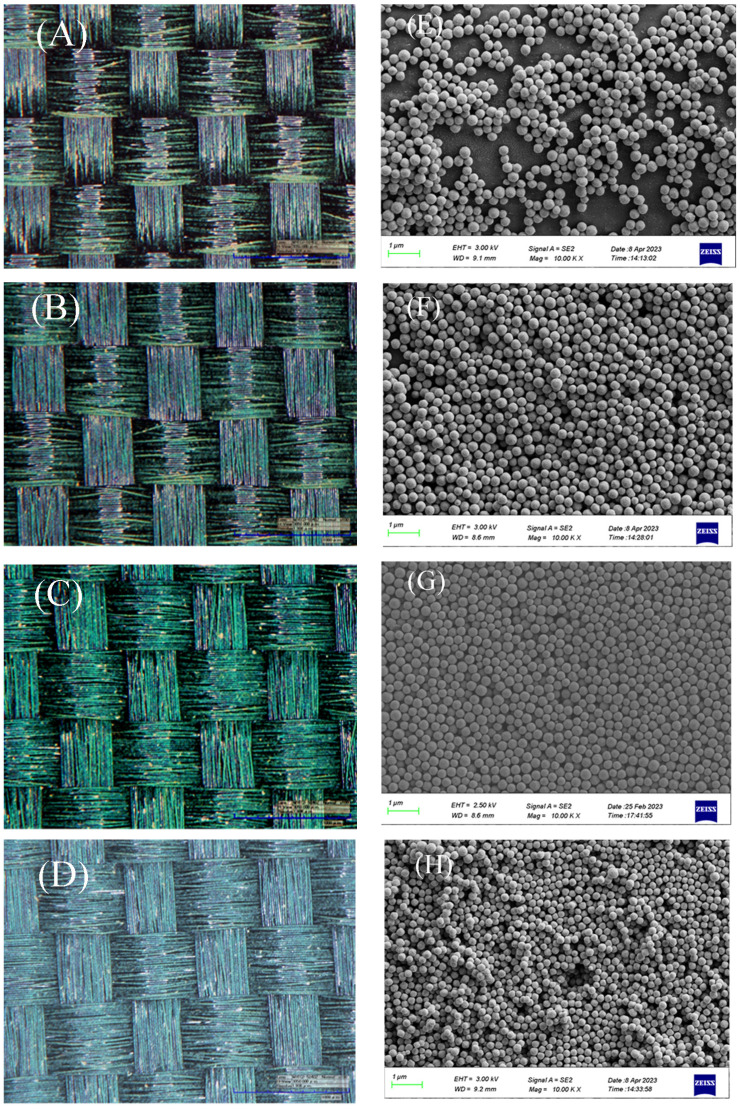
The structural color of fabrics with different Cu_2_O microsphere concentration (**A**) 0.5 wt.%; (**B**) 1.0 wt.%; (**C**) 3.0 wt.%; (**D**) 5.0 wt.%; (**E**–**H**) the SEM images of Cu_2_O microspheres.

**Figure 8 materials-17-03238-f008:**
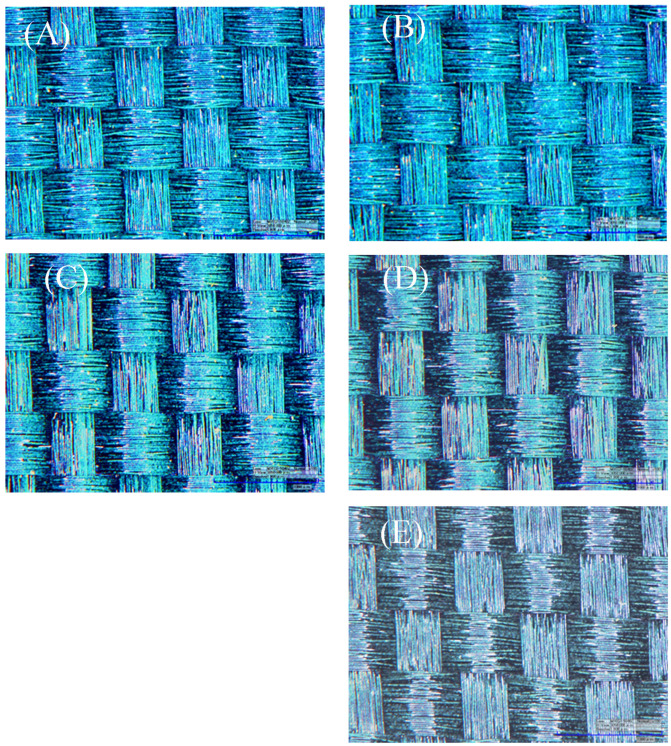
The effect of concentration of poly(butyl acrylate) on structural color. (**A**) 5.0 wt.%; (**B**) 10.0 wt.%; (**C**) 15.0 wt.%; (**D**) 20.0 wt.%; (**E**) 25.0 wt.%.

**Figure 9 materials-17-03238-f009:**
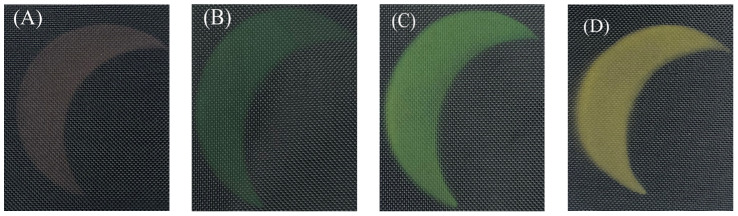
Structural color from Cu_2_O microsphere with different sizes (**A**) 190 nm (**B**) 210 nm (**C**) 240 nm (**D**) 275 nm.

**Figure 10 materials-17-03238-f010:**
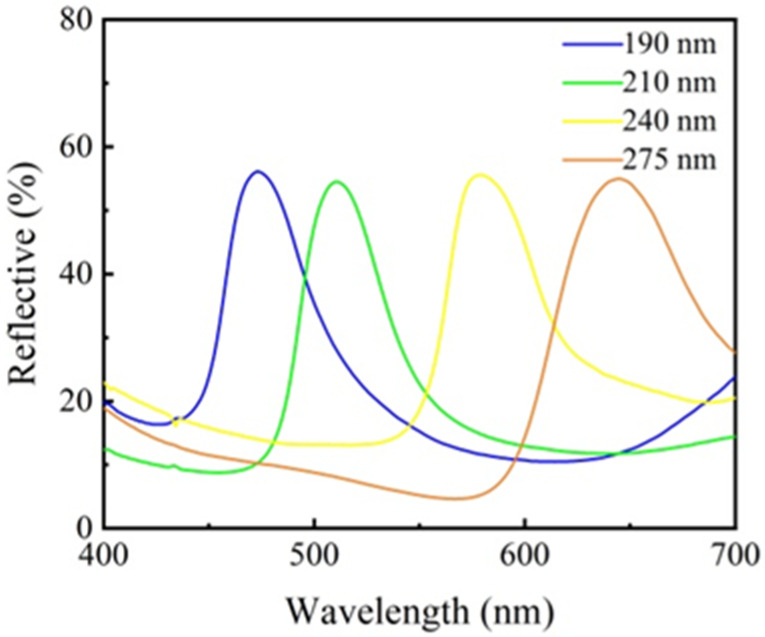
Reflectance curves of structural colored fabrics by adding 5.0 wt.% poly(butyl acrylate) to Cu_2_O microspheres with different particle sizes, i.e., 275 nm, 240 nm, 210 nm, and 190 nm.

**Figure 11 materials-17-03238-f011:**
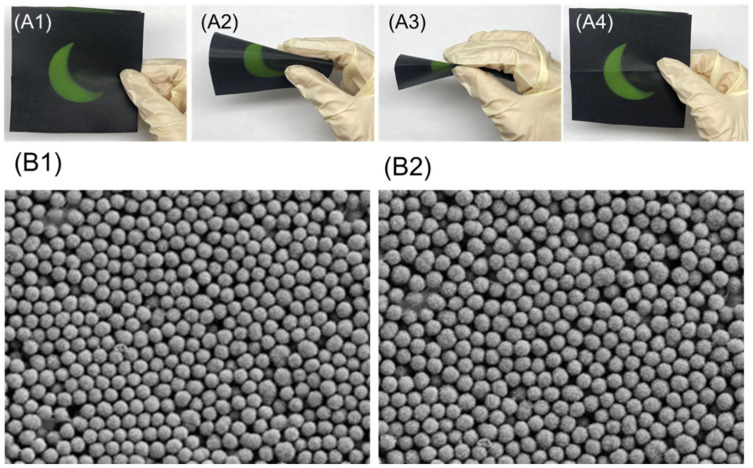
Bending test of structural colored fabric, (**A1**–**A4**) Bending test process. (**B1**) SEM image of photonic crystal before bending test; (**B2**) SEM image of photonic crystal after bending test.

**Figure 12 materials-17-03238-f012:**
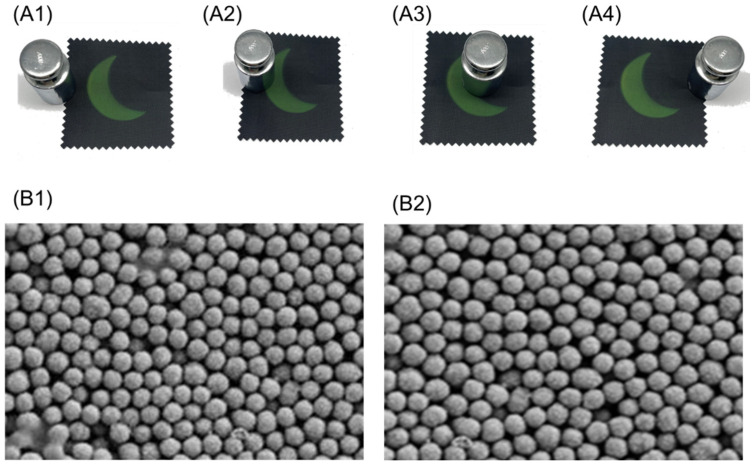
Friction test of structural colored fabric, (**A1**–**A4**) friction test process; (**B1**) SEM image of photonic crystal before friction test; (**B2**) SEM image of photonic crystal after friction test.

**Table 1 materials-17-03238-t001:** Diameter of Cu_2_O microsphere and Zeta potential value.

Sample	Diameter of Cu_2_O Microsphere (nm)	Zeta Potential (mV)
Cu_2_O microsphere dispersions	275	−31.24
	240	−32.34
	210	−30.85
	190	−33.43

## Data Availability

Data is contained within the article.

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
