# Peer review of "Structural Colored Fabric Based on Monodisperse Cu2O Microspheres"

_materials, 2024, doi:10.3390/ma17133238_

Round 1

Reviewer 1 Report

Comments and Suggestions for Authors

The paper reports on the preparation of copper oxide microspheres and the study of fabrics coated with these microspheres. The subject of the paper fits well the scope of Materials journal.

I have the following comments:

1. In the Introduction section, the second paragraph makes no sense from a chemical point of view. Please re-write this paragraph.

2. Similarly, the first sentence in the last paragraph of the Introduction section also makes no sense.

3. In the Materials and Methods section, the information on the molecular weight of polyvinylpyrrolidone is absent.

4. In the Section 2.1, please provide the information on poly(butyl acrylate) used, including the information on its molecular weight.

5. Throughout the manuscript, copper oxide microspheres are referred to as single crystals. Please provide any experimental confirmation on the single crystal nature of the microspheres (e.g. electron diifraction data etc.).

6. What is the reason for the increase in the size of microspheres upon storage? Please explain. Please also provide an independent conformation of such an increase using e.g. SEM data.

Comments on the Quality of English Language

Moderate editing of English language is required.

Reviewer 2 Report

Comments and Suggestions for Authors

The paper “Structural colored fabric based on monodisperse single-crystal Cu2O microspheres” proposed for the journal Materials, is suitable for the chosen journal.

 The abstract synthesizes well the content of the paper.

As general comment, I recommend a revision of the English language because there some unclear sentences and some errors.

 I have some specific comments for the paper.

 1) page 2, lines 50-52: Later, they [21] extended their study by studying distinct colors obtained on transparent substrates through the structural arrangement of these single-crystal Cu2O spheres.

Please revise the sentence (their study by studying …

Number 2 in the chemical formula must be subscript.

 2) page 2, lines 64-65: Despite some preliminary research work has been done to explore the potential of Cu2O in the structural color of textiles.

The verb is lacking, the authors must add the end of the sentence, it is incomplete.

 3) page 2, line 65: This research …

Which one? It is not clear what the authors would like to say. To which research they are referring?

 4) page 3, lines 120-121: The hydrated size and monodispersity of the Cu2O microsphere were analyzed by the dynamic light scattering and zeta potential measurement systems.

Please remove this sentence, because it is a repetition (the techniques have been reported at the beginning of the sub-paragraph).

 5) Page 3, line 124: please specify why did you use FTIR.

 6) page 4, line 148: (KH-7700) from QUESTAR China Limited)

Please check the round brackets.

 7) page 4, lines 151-153: this part is confused. Bending and friction test must be described and it should be explained their use.

 8) page 4, subparagraph 3.1: according to my opinion this part should be moved to the Materials and Methods section. This is not a result. Moreover, the concept expressed at lines 170-171 (i.e. it is required that the monodispersity index of the prepared microspheres be less than 0.08) has been reported previously at line 118.

 9) page 5, figure 1: please specify that the images are from FSEM. Why the measure unit is not reported in the images over the scale bar?

 10) page 7, caption of figure 5: please check the subscript in the chemical formula of copper oxide. The same in the caption of figure7.

 11) page 7, lines 237-238: Usually, when the absolute value of the Zeta potential of particles in the aqueous phase is greater than or equal to 30 mV, it is considered that the dispersion stability of the system is good.

Could the authors add references for this assessment? It is considered that … by which scholars?

 12) Page 8: samples 1, 2, 3 and 4. what are these samples? Never in the text was there any reference to the samples with the numbers 1,2, 3 and 4.

 13)  Page 8, lines 272-273: Due to the high concentration of Cu2O microspheres obtained by centrifugation, they are prone to adhesion and cannot be directly applied to the spray preparation of structural color-printed patterns.

This is not clear for me, please explain the assessment.

 14) Page 8, line 278: 0.5 w.%

Correct with 0.5 wt.%

 15) Page 9: in the figure 7 the authors have to add description also of images on the right side of the figure even if they are clearly from scanning electron microscope.

 16) page 9, line 290: Figure 7 e-h for example. The arrangement of particles is not shown in 7a-d but on the right side FSEM images.

 17) page 10, line 295: Figure 7e

 18) Please remove the sentence: The microsphere arrangement of the photonic crystal structure obtained above was characterized, and the results are shown in Figure 7.

It has no sense; you have introduced Figure 7 before.

 19) At the end of the page 9 you write: The arrangement of Cu2O microspheres was observed as well and given in Figure 7 (a-d). The concentration of Cu2O microspheres had an effect on their arrangement.

 And later, at page 10, lines 297-298: It can be seen from the figure that the concentration of microspheres will affect the arrangement of microspheres to some extent.

 Please revise this part, it is confused.

 20) Page 11, lines 328-329: Study the effect of different binder ratios on the printing effect of Cu2O microsphere structural color patterns. Observe the surface morphology of the obtained structural color pattern and the arrangement of microspheres inside it.

I don’t understand the sense of these two sentences. There is no verbs. Are they titles of subparagraphs?

Please check and correct. In the text at page 11 I suggest to add that the binder is poly(butyl acrylate).

 21) Page 11-12, paragraph 3.9. The result shown in the figure 9 must be explained and the consequent discussion must be improved.

 22)  Page 13: please explain the friction test and how did you evaluate the result.

The text in the paragraph 3.10.2 must be re-organized and revised.

Befofe you write: The structural colored fabrics demonstrated the same color before and after the friction test (Figure 12 A1-A4), which indicated that the Cu2O photonic crystal had a stable structure. The SEM images of Cu2O photonic crystal before and after the friction test showed the orderly arrangement, which also approved it. What do you mean with this last words?

Then you write: A friction experiment was conducted on the stability of the structural color pattern. The results are shown in Figure 12

You presented again the test and the figure 12 already shown before, at the beginning of the sub-paragraph.

 23) lines 388-389: It can be seen that after the Cu2O photonic crystal structure was rubbed, the structural color pattern did not fall off significantly.

 Where this is visible? It is not clear.

 24) page 13, lines 390-394: The butyl ester adhesive enhances the bonding strength between the Cu2O photonic crystal and the substrate, and the microspheres in the FESEM image are tightly arranged and orderly. This shows that the addition of poly(butyl acrylate) enhances the stability between the Cu2O photonic crystal color structure and the substrate and does not easily fall off from the substrate during friction, indicating that its color structure has excellent durability.

 This part appears confused and must be revised. I don’t understand as I can see how the addition of poly-BA enhances the stability of the system.

 In the conclusion the authors must report the most relevant results in synthesis, and they should spend some words to discuss about future possible development of their research.

 References must be checked and formatted according to the standard of the journal Materials.

Comments on the Quality of English Language

A revision of the English language is requested because there some unclear sentences and some errors.

Reviewer 3 Report

Comments and Suggestions for Authors

Dear Authors,

the development of alternatives of current dyes used in textile industry is interesting and need more research to understand and explain the mechanisms.

I have some remarks and questions about your paper.

The first remarks concerns the form of this paper. At least two times paragraphs are completely repeated (lines 212-222 = line 202-212 ; lines 245-248 = line 235-239]. Modification of the font size (lines 279-286). 

I suggest to plot the two curves of the figure 5 on the same graph. Please revise also the title of this figure.

Based on the Bragg law you explained how was found the target of microsphere's diameter. In the paper the diameter obtained with the different ratios studied are in the targeted domain. However how have you determined the molar ratio of citrate to Cu2+ at the beginning of the study?

Furthermore how have you determined the concentration of microspheres and concentration of poly(butyl acrylate). Could you explain the choice of this binder according the substrate (polyester woven fabric) and the requirements?

Line 184 :"monodispersed indexes. which indicated....excellent stability and monodispersity". Based on this index how can you concluded about the stability?

Line 230:"greater than or equal to 30 mV..." Please add a reference to justify this.

In figure 6B you have erratic variations. Have you an explanation about this or these variations are not significant?

Could you give more details about the parameters used during the spraying? In general you should clearly indicate all the details for each sample tested (ex: fig 7: which size of microsphere is used? fig 11: method used for application of microspheres, concentration?)

You should add a figure with the raw woven fabric to have an idea of its color.

K/S indicator or data about the CIELAB space should be added to comment more deeply the color variations.

Some conclusions shoud be completed with data. ex: line 300 "large spacing..." ==> give the averaged distance, line 302 :"the fabric cannot be completely covered" ==> a "coverage" factor can be calculated with image treatment, line 342 :"are arranged in an orderly manner" ==> have you SEM images to support this conclusion?

Figure 9 and 10 : could you comment (aind explain) if the color obtained and  the spectra recorded are in accordance with the expected results due to the size of microspheres.

Round 2

Reviewer 1 Report

Comments and Suggestions for Authors

The authors have addressed all of my comments. The paper was improved substantially. Thus I would like to recommend it for publication in Materials.

Comments on the Quality of English Language

 Minor editing of English language is required

Reviewer 2 Report

Comments and Suggestions for Authors

I checked the revised version of the paper and I found that the authors improved significantly the manuscrpt.

I think that now it can be accepted for publication